# Sensory experience modifies feature map relationships in visual cortex

**Shaun L Cloherty**[1,2,3†§], **Nicholas J Hughes**[4,5†], **Markus A Hietanen**[1,2], **Partha S Bhagavatula**[1,2], **Geoffrey J Goodhill**[4,5*‡], **Michael R Ibbotson**[1,2*‡]

[1]National Vision Research Institute, Australian College of Optometry, Carlton, Australia; [2]ARC Center of Excellence for Integrative Brain Function, Department of Optometry and Vision Sciences, University of Melbourne, Parkville, Australia; [3]Department of Electrical and Electronic Engineering, University of Melbourne, Parkville, Australia; [4]Queensland Brain Institute, The University of Queensland, St Lucia, Australia; [5]School of Mathematics and Physics, The University of Queensland, St Lucia, Australia

**\*For correspondence:**
g.goodhill@uq.edu.au (GJG);
mibbotson@nvri.org.au (MRI)

[†]These authors also contributed equally to this work
[‡]These authors also contributed equally to this work

**Present address:** [§]Department of Physiology, Monash University, Clayton, Australia

**Competing interests:** The authors declare that no competing interests exist.

**Abstract** The extent to which brain structure is influenced by sensory input during development is a critical but controversial question. A paradigmatic system for studying this is the mammalian visual cortex. Maps of orientation preference (OP) and ocular dominance (OD) in the primary visual cortex of ferrets, cats and monkeys can be individually changed by altered visual input. However, the spatial relationship between OP and OD maps has appeared immutable. Using a computational model we predicted that biasing the visual input to orthogonal orientation in the two eyes should cause a shift of OP pinwheels towards the border of OD columns. We then confirmed this prediction by rearing cats wearing orthogonally oriented cylindrical lenses over each eye. Thus, the spatial relationship between OP and OD maps can be modified by visual experience, revealing a previously unknown degree of brain plasticity in response to sensory input.

## Introduction

In cats and monkeys neurons in the primary visual cortices are selective for both the orientation of the visual input (orientation preference, OP) and its eye of origin (ocular dominance, OD) (*Hubel and Wiesel, 1977*). These feature preferences are arranged spatially in the form of OP and OD maps, with stereotypical structure within each map, and strong spatial relationships between them (*Blasdel and Salama, 1986*; *Bonhoeffer and Grinvald, 1991*; *Bartfeld and Grinvald, 1992*; *Obermayer and Blasdel, 1993*; *Hübener et al., 1997*; *Nauhaus et al., 2012*). While some aspects of the structure of OD and OP maps individually are plastic in response to altered visual input, such as monocular deprivation (*Hubel et al., 1977*; *Shatz and Stryker, 1978*; *Farley et al., 2007*) or stripe rearing (*Sengpiel et al., 1999*; *Tanaka et al., 2006*), none of these manipulations has succeeded in modifying the overall spatial relationships between OD and OP maps, which have appeared immune from environmental influence. In particular OP map pinwheels, where domains representing all orientations meet at a point, always tend to lie close to the center of OD regions. This is true even after manipulations of the visual input such as rearing animals with artificially induced strabismus (*Hubel and Wiesel, 1965*; *Löwel, 1994*; *Löwel et al., 1998*) or monocular deprivation (*Crair et al., 1997*). However, whether this relationship is a fundamental aspect of map structure that is determined by innate mechanisms (*Godecke and Bonhoeffer, 1996*; *Crair et al., 1998*; *Kaschube et al., 2002*; *Katz and Crowley, 2002*; *Tomita et al., 2013*), and thus beyond the limits of brain plasticity, is unclear.

**eLife digest** The structure of the brain results from a combination of nature (genes) and nurture (environment). The brain's ability to adapt to changes in the environment is known as plasticity, and the young brain is especially plastic. An animal's sensory experiences in early life help to determine how its brain will process sensory input as an adult. One of the best sensory systems in which to study this process is the visual system.

Within the visual system, some brain cells respond only to input from the left eye and others only to input from the right eye. Cells that respond to input from the same eye are arranged to form columns. Within each column, some cells respond only to lines with a particular orientation. Cells with different preferred orientations are grouped together in patterns that resemble pinwheels. The relative positions of the pinwheels and eye-specific columns within the brain tissue belonging to the visual system have so far been robust to changes in visual experience during development, suggesting that they are determined by an animal's genes.

However, Cloherty, Hughes et al. have now tested the unexpected predictions of a computer model. The model suggested that rearing animals so that they saw mostly vertical lines through one eye, and mostly horizontal lines through the other, would cause a form of plasticity that had never been observed before. Specifically, it would change the relative positions of the pinwheels and eye-specific columns within the visual parts of the brain. This prediction turned out to be correct. Young cats that wore special lenses – which slightly distorted what they saw but did not obviously affect their behavior – showed the predicted changes in brain structure.

The results confirm that this aspect of brain structure is partly determined by nurture, as opposed to being entirely specified by nature. A key future challenge is to identify the chemical signaling that enables sensory input to have these effects on brain structure. It might then be possible to use drugs to restore normal brain activity in cases where abnormal sensory input has altered the brain, for example in the condition known as amblyopia (or "lazy eye").

Computational models of map formation based on Hebbian plasticity principles have played an important role in understanding the mechanisms governing visual development. In particular 'dimension reduction' models predict negative correlations between the local gradient magnitudes of different maps (*Durbin and Mitchison, 1990*; *Swindale, 1996*), thus explaining why pinwheels, which have a high orientation gradient, normally tend to lie near the centre of OD regions, where the ocularity gradient is small. However, such models also suggest that the spatial relationship between pinwheels and OD regions might be sensitive to visual experience (*Giacomantonio et al., 2010*). Here we used a computational model to predict how rearing animals with visual input biased to vertical orientations in one eye and horizontal orientations in the other eye (cross-rearing) would change these relationships. We then confirmed this prediction by raising cats with weak (-10 dioptre) cylindrical lenses placed in front of their eyes throughout the critical period. This demonstrates a form of plasticity in the relationships between visual feature maps that has not previously been observed.

## Results

### Computational prediction

The elastic net algorithm (*Durbin and Mitchison, 1990*) uses Hebbian learning to optimize a trade-off between coverage and continuity constraints, and can explain many aspects of visual map formation (*Swindale, 1996*; *Goodhill, 2007*). When simulating normal rearing (*Erwin et al., 1995*; *Carreira-Perpinan and Goodhill, 2004*; *Carreira-Perpinan et al., 2005*) this reproduces the experimental observations cited above that pinwheels tend to be located near the center of OD columns. However, previous simulations of map development (*Giacomantonio et al., 2010*) suggested that this relationship would be disrupted when horizontal orientations were over-represented in one eye and vertical orientations were over-represented in the other (*Hirsch and Spinelli, 1970*; *1971*; *Blakemore, 1976*). Here we simulated this scenario using the elastic net algorithm to quantify

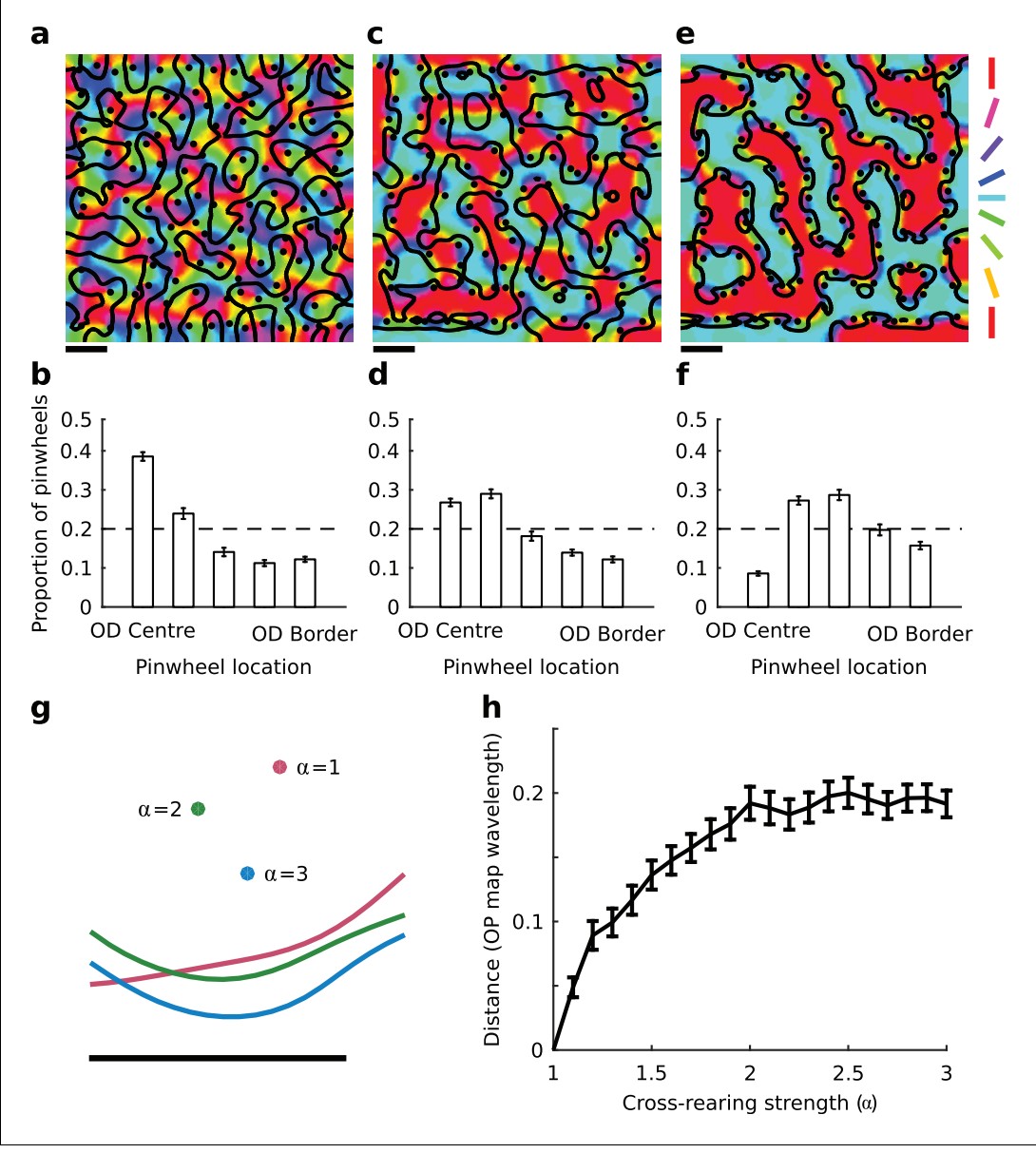

**Figure 1.** The elastic net model predicts changes in spatial map relationships under cross-rearing. (a) Simulated orientation preference map (colours), orientation pinwheels (black dots), and ocular dominance borders (black lines) under normal rearing (relative strength of over-representation of horizontal and vertical contours in the input $\alpha = 1$). (b) Histogram of pinwheel locations relative to the OD borders under normal rearing, showing a preference for pinwheels located near the centre of OD regions as previously observed experimentally. Error bars show ± 1 SEM from 10 independent simulations. The dashed line shows the expected distribution if pinwheels were arranged randomly. (c) Simulated orientation preference map for the cross-reared condition ($\alpha = 3$). (d) Histogram of pinwheel locations relative to OD borders for this case. (e,f) Simulated orientation preference maps and corresponding histogram of pinwheel locations relative to OD borders for a higher level of cross-rearing ($\alpha = 5$). In the simulations of cross-rearing, pinwheels are shifted away from the centre and towards the border of OD regions. (g) A cropped region of a simulated OP and OD map produced with the same random seed but increasing strengths of over-representation. Circles show the location of a pinwheel and lines show the location of the adjacent OD border. Both the pinwheel and the OD border move under cross-rearing relative to their positions under normal rearing ($\alpha = 1$), but the distance between them decreases. (h) The average distance that pinwheels move from their original positions (measured in units of average OP map wavelength) as a function of the strength of cross-rearing. Errors bars show ± 1 SEM across all pinwheels in the map. Scale bars in panels **a, c, e** and **g** indicate 15 pixels in the simulated feature maps. Source data for this figure are available in *Figure 1— source data 1*.

*Figure 1 continued on next page*

*Figure 1 continued*

The following source data is available for figure 1:

**Source data 1.** This HDF5 file contains the numerical values shown in *Figure 1*.

further the degree to which the relationship between pinwheels and OD borders would change as a function of the strength of over-representation (see Materials and methods).

To measure the relationship of pinwheels to OD regions we separately divided the left and right eye regions of simulated OD maps into five equally sized bins, which represented areas of the OD map from the centres of the OD columns to the border regions, similarly to *Hübener et al. (1997)*. As the strength of over-representation (α) increased, the histograms became increasingly biased away from the center of OD regions, indicating a shift in the relative location of pinwheels towards the OD borders (*Figure 1*). To determine whether this is due to the movement of pinwheels, OD borders, or both, we then fixed the random seed in the algorithm and explored how the spatial relationships changed within reproducible maps as a function of α. An example is shown in *Figure 1g*, from which it is clear that both pinwheel positions and OD borders move in the cross-reared case. The average distance that pinwheels move from their original (α = 1) positions as a function of α is shown in *Figure 1h*. Thus, in the model, cross-rearing alters the spatial relationship between pinwheels and OD borders by causing movement of both.

However, plasticity of this type has not yet been observed experimentally, leaving open the possibility that the relationship between pinwheels and OD columns could instead be determined by intrinsic mechanisms and is not susceptible to environmental modification.

## Testing the prediction

To directly test these predictions we reared cats from 3 weeks of age with -10 dioptre cylindrical lenses mounted comfortably in front of their eyes using soft neoprene masks. The lens covering the left eye had its axis aligned vertically, so that the left eye was exposed to high contrast contours with primarily horizontal orientation. Conversely, the lens covering the right eye had its axis aligned horizontally, so that the right eye was exposed to high contrast contours with primarily vertical orientation (*Figure 2*). The lens strength was chosen based on preliminary observations that animals wearing -10 dioptre lenses exhibited normal behavioural activity, while animals with higher power lenses (e.g., as used in *Tanaka et al. [2006]*) were noticeably less active. Animals wore the masks for 6 hr per day while the room was illuminated and were otherwise kept in darkness. During light periods the animals were monitored at least every 30 min to ensure the masks remained in place and to encourage active visual behaviours, such as chasing light patterns and balls. Paw striking behaviour towards objects in front of the animals appeared normal. From age 20 weeks we used intrinsic signal optical imaging followed by extracellular single unit recordings to map OP and OD preferences in cortical areas 17 and 18 of 5 cross-reared animals (10 hemispheres), and 6 normally reared control animals (11 hemispheres).

## Cross-rearing alters tuning properties of single units

We recorded extracellular spiking responses to quantify the tuning properties of single units in both the normal and cross-reared animals. In the five cross-reared animals we recorded from 182 units from 20 electrode tracks, 76 with a preference for input from the right eye (which experienced predominantly vertical contours) and 106 with a preference for input from the left eye (which experienced predominantly horizontal contours). In the six control animals, we recorded from 86 units from 17 electrode tracks. Electrode tracks were positioned without reference to cortical map structure.

Consistent with previous work (*Coppola et al., 1998*; *Li et al., 2003*) and theoretical predictions (*Hunt et al., 2013*), the distribution of preferred orientation for units from the control animals exhibited an over representation of the cardinal orientations (*Figure 3a*). This distribution was well described by a sine curve with period 90° ($r^2$ = 0.69). In contrast, distributions of preferred orientation for units from the cross-reared animals showed clear biases depending on the eye providing the dominant input. Units with a preference for input from the left eye showed a bias for horizontal

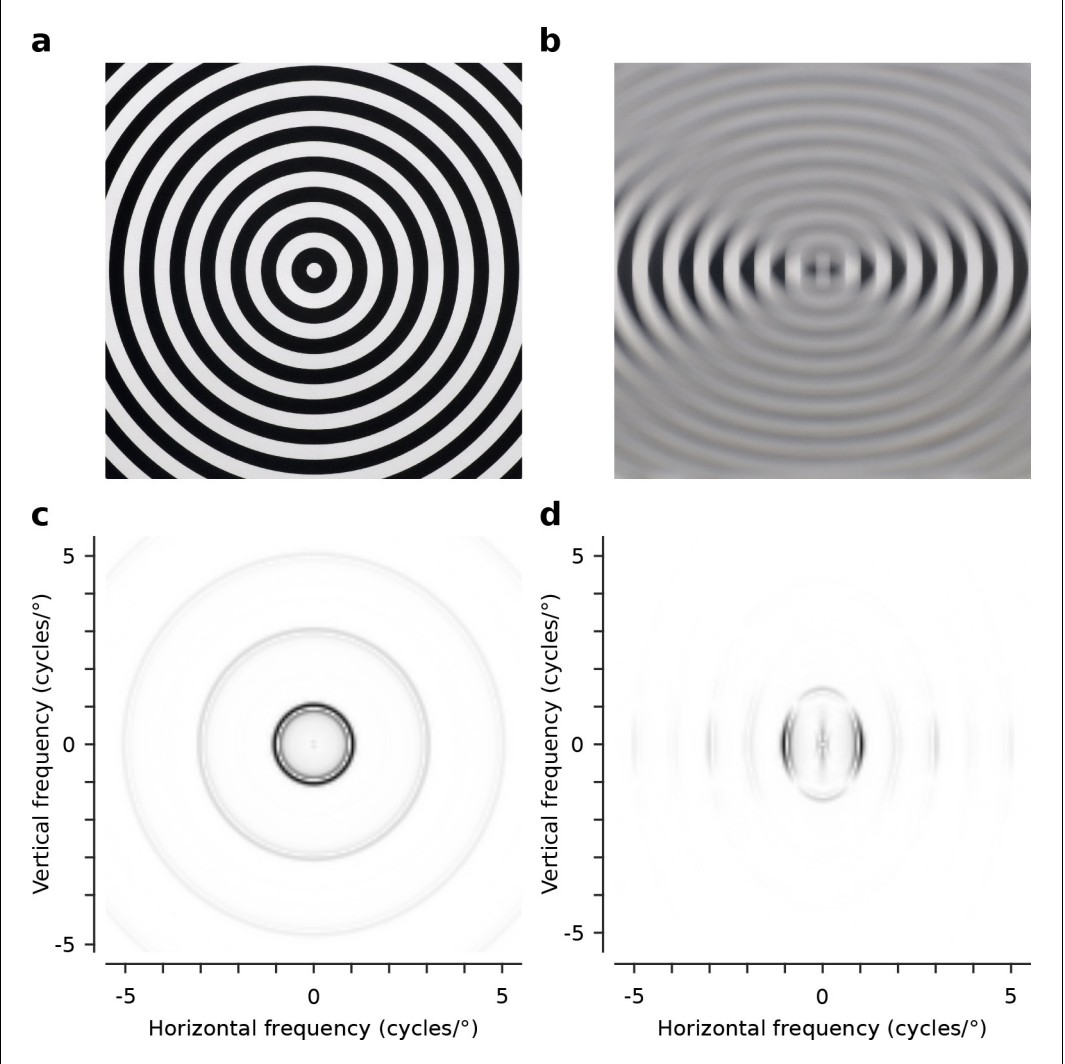

**Figure 2.** Optical characteristics of the -10 dioptre cylindrical lenses. (**a**) Circular square wave test grating (1 cycle/°) viewed normally (no lens). (**b**) The same grating viewed through the -10 dioptre cylindrical lens, with the lens axis aligned horizontally, attenuating horizontal and preserving vertical contours. (**c**) The radially symmetric distribution of power over spatial frequency for the test grating viewed normally. (**d**) The radially asymmetric distribution of power over spatial frequency for the same grating when viewed through the -10 dioptre cylindrical lens with the lens axis aligned horizontally. Contours orthogonal to the axis are preserved while contours parallel to the axis are attenuated. Power spectra shown in (**c**) and (**d**) are normalized to the peak power (black).

orientations (*Figure 3b*, black line), while those with a preference for input from the right eye showed a bias for near vertical orientations (*Figure 3b*, gray line). These distributions were consistent with the orientation of the lenses fitted over each eye and were well described by sine curves with period 180° ($r^2 = 0.75$, left eye; $r^2 = 0.79$, right eye). In contrast to the distribution from the control animals, sine curves with period 90° provided a relatively poor account of the data ($r^2 = 0.11$, left eye; $r^2 = 0.03$, right eye).

We also calculated the inter-ocular difference in preferred orientation, ΔOP, for each unit as the preferred orientation for the right eye minus the preferred orientation for the left eye. Consistent with previous reports (*Nelson et al., 1977*; *Cooper and Pettigrew, 1979*) we found a torsional disparity in the preferred orientation of the two eyes in both our control (mean ΔOP = 11.7°, p<0.001, two-tailed t-test) and cross-reared (mean ΔOP = 9.8°, p<0.001, two-tailed t-test) animals. We found no significant difference in the distribution of ΔOP for control and cross-reared animals (p=0.44,

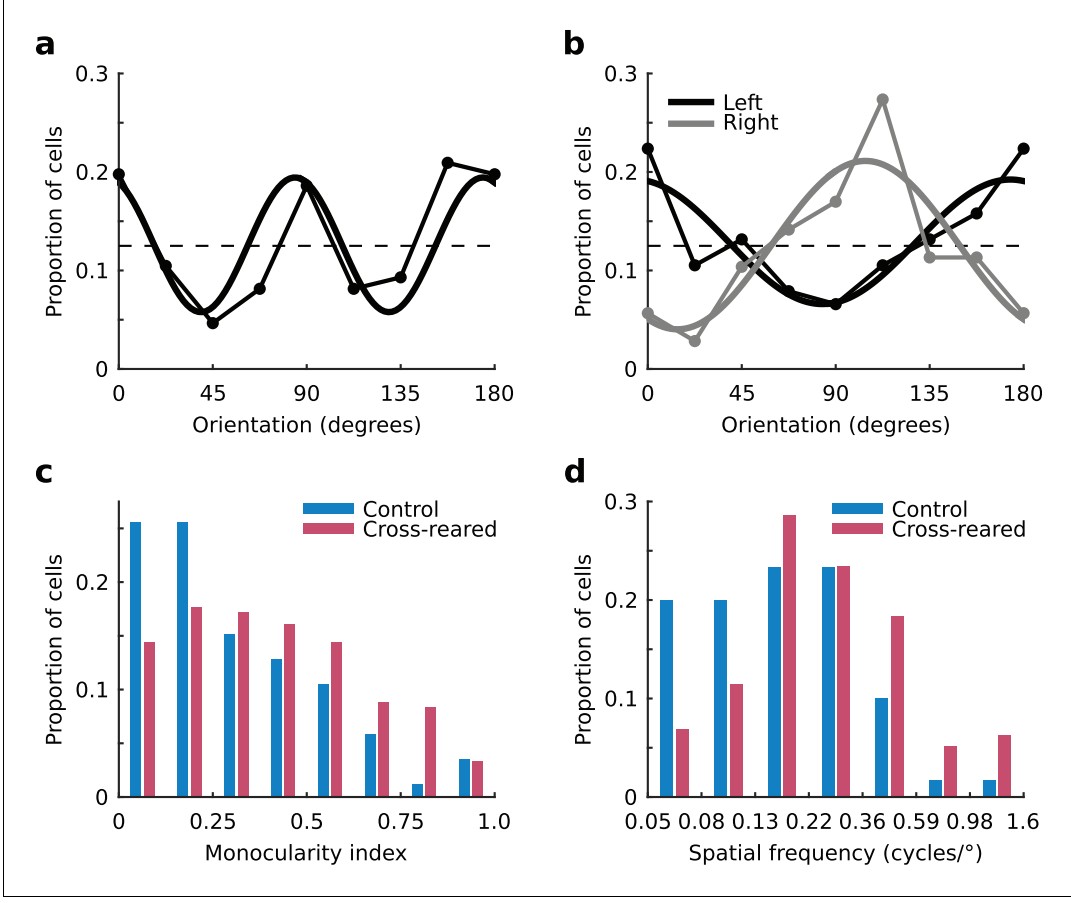

**Figure 3.** Tuning properties of single units. (**a**) The distribution of preferred orientation in control animals exhibited an over representation of cardinal orientations. The dashed line shows the expected distribution if all orientations were represented equally. The best-fitting sine curve with period 90° had peaks at 84° and 174° (thick line, $r^2$ = 0.69). (**b**) Distributions of preferred orientation in the cross-reared animals, for units driven predominantly by input from the left (black) and right eye (grey). The best-fitting sine curves with period 180° peaked at 174° for the left eye (thick black line, $r^2$ = 0.75) and 104° for the right eye (thick grey line, $r^2$ = 0.79). Cross-rearing thus caused a systematic change in the distributions of preferred orientation of single units. Cross-rearing also caused an increase in monocularity (**c**) and an increase in preferred spatial frequency (**d**) of single units. Source data for this figure are available in **Figure 3—source data 1**.

The following source data and figure supplements are available for figure 3:

**Source data 1.** This HDF5 file contains the numerical values shown in **Figure 3**.

**Figure supplement 1.** Inter-ocular difference in preferred orientation of single units is not altered by cross-rearing.

**Figure supplement 2.** Preferred temporal frequency of single units is not altered by cross-rearing.

**Figure supplement 3.** Contrast sensitivity of single units is not altered by cross-rearing.

two-tailed, two-sample t-test; **Figure 3—figure supplement 1**). Units from cross-reared animals showed a higher level of monocularity than those from control animals (**Figure 3c**). The median monocularity index (*MI*, see Material and methods) of units from control and cross-reared animals was 0.24 and 0.38, respectively. This difference was significant (p=0.002, Kruskal-Wallis test). Cross-rearing therefore appears to induce subtle changes in the combination of input from the two eyes at the level of single neurons.

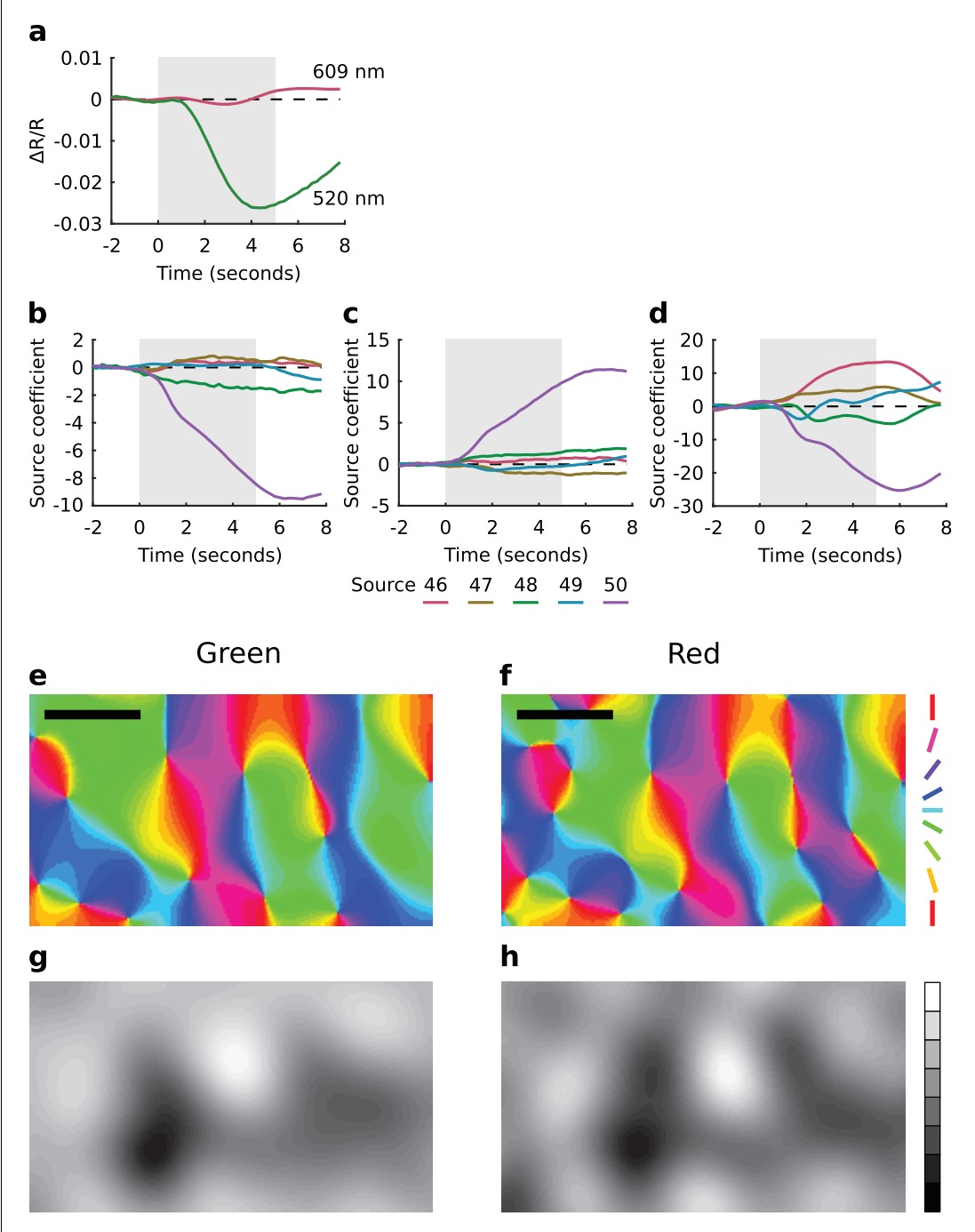

**Figure 4.** Extended spatial decorrelation recovers OP and OD maps from green light imaging. (a) Time course of the relative change in reflectance (ΔR/R) during a trial, averaged over all pixels and all trials, measured with red (609 nm) and green (520 nm) light. The shaded region shows the stimulus period. Consistent with earlier reports (**Sirotin and Das, 2009**; **Sirotin et al., 2009**), green light produced a much stronger signal. (b–d) Representative source coefficient time series from the extended spatial decorrelation algorithm (see Materials and methods). Sources 46–50 (as per legend) for (b) the real component of the OP map, (c) the imaginary component of the OP map, and (d) the OD map. The shaded region shows the stimulus period. The mean over the pre-stimulus period was subtracted from each source. It is clear in each case that one source (in this case source 50) most strongly represents the signal of interest. (e,f) OP maps generated from the green (e) and red (f) light responses. (g,h) OD maps generated from the green (g) and red (h) light responses. Over the region shown here (the one used for analysis), the correlation between red and green OP maps was $r^2 = 0.78$ and between the red and green OD maps was $r^2 = 0.77$. Colour encodes the preferred orientation in the OP maps (as per legend) and brightness encodes eye preference in the OD maps, with black and white representing the left and right eyes, respectively. Data from a control animal. Scale bars: 1 mm. Source data for this figure are available in **Figure 4—source data 1**.

*Figure 4 continued on next page*

*Figure 4 continued*

The following source data is available for figure 4:

**Source data 1.** This HDF5 file contains the numerical values shown in *Figure 4*.

In addition to each unit's ocular dominance and orientation preference, we also quantitatively measured their tuning for spatial and temporal frequency (See Materials and methods). We found no significant differences in the monocular spatial or temporal frequency tuning for dominant vs non-dominant eyes or for left vs right eyes (regardless of dominance) in either control or cross-reared animals ($p > 0.05$, Kruskal-Wallis tests). We therefore combined the populations of left- and right-eye dominant units within the two experimental groups (i.e., control and cross-reared) and compared their tuning parameters for the dominant eye. Units from cross-reared animals exhibited marginally higher preferred spatial frequencies compared to those from control animals (median preferred spatial frequency 0.24 and 0.16 cycles/° for cross-reared and control animals respectively; *Figure 3d*). While this difference was significant ($p < 0.001$; Kruskal-Wallis test), the tuning curves were very broad and overlapped substantially. We found no difference in either the bandwidth ($p = 0.19$; Kruskal-Wallis test) or skew ($p = 0.40$; Kruskal-Wallis test) of the spatial frequency tuning curves for normal and cross-reared animals. Similarly, we found no significant difference in either the preferred temporal frequency ($p = 0.2$, Kruskal-Wallis test; *Figure 3—figure supplement 2*), the temporal frequency bandwidth ($p = 0.28$, Kruskal-Wallis test) or the skew of the temporal frequency tuning curves ($p = 0.63$, Kruskal-Wallis test) of units from control and cross-reared animals.

Using the optimal grating parameters (orientation, size, spatial and temporal frequency) for each unit we also measured their response as a function of stimulus contrast. We found no difference in either the maximum spike rate ($p = 0.56$, Kruskal-Wallis test) or the semi-saturation contrast ($p = 0.81$, Kruskal-Wallis test; *Figure 3—figure supplement 3*) of units from control and cross-reared animals. Selectivity for stimulus parameters, responsiveness and sensitivity to stimulus contrast therefore appeared normal in the cross-reared animals.

## Extended spatial decorrelation for intrinsic signal optical imaging

While previous studies have usually used red light (wavelengths >600 nm) for intrinsic signal imaging, here we used green light (520 nm) due to the much higher signal to noise ratio obtained (*Figure 4a*). Conventional techniques for map generation from intrinsic signal imaging data are based on the calculation of difference images by either subtraction or division by a reference image that is independent of the stimulus of interest. We found that these techniques left strong blood vessel artifacts in the maps derived from green light data, particularly for ocular dominance maps. However we were able to overcome this problem by using extended spatial decorrelation (ESD), a more sophisticated analysis technique (*Stetter et al., 2000*) (see Materials and methods). This technique robustly separated the noise and mapping signals even in the green light data (*Figure 4b–d*). Consistent with a brief earlier report (*Frostig et al., 1990*), by imaging one of our control animals with both green (520 nm) and red (609 nm) wavelengths we found that the maps produced were very similar (*Figure 4e–h*). We also compared measures of orientation preference from the OP maps with corresponding measures obtained from single unit recordings from the superficial layers of the cortex at each electrode track location. Where we obtained robust estimates from both the imaging and unit recordings we found a high level of correlation between the two measures ($r^2 = 0.64$, $p = 0.003$) with a median absolute difference of 15.2°.

## Cross-rearing alters the proportion of cortical area representing different orientations

We measured OP and OD maps in cortical areas 17 and 18 of both normal and cross-reared animals using intrinsic signal optical imaging (typical maps for each case are shown in *Figure 5a–f*). Cross-rearing caused a very slight reduction in orientation selectivity, as derived from the OP maps (*Figure 5—figure supplement 1*). Although the selectivity distributions appear very similar, the difference was statistically significant ($p < 0.001$, two-sample Kruskal-Wallis test). However cross-rearing induced profound changes in map structure. In control animals there was an over-representation of

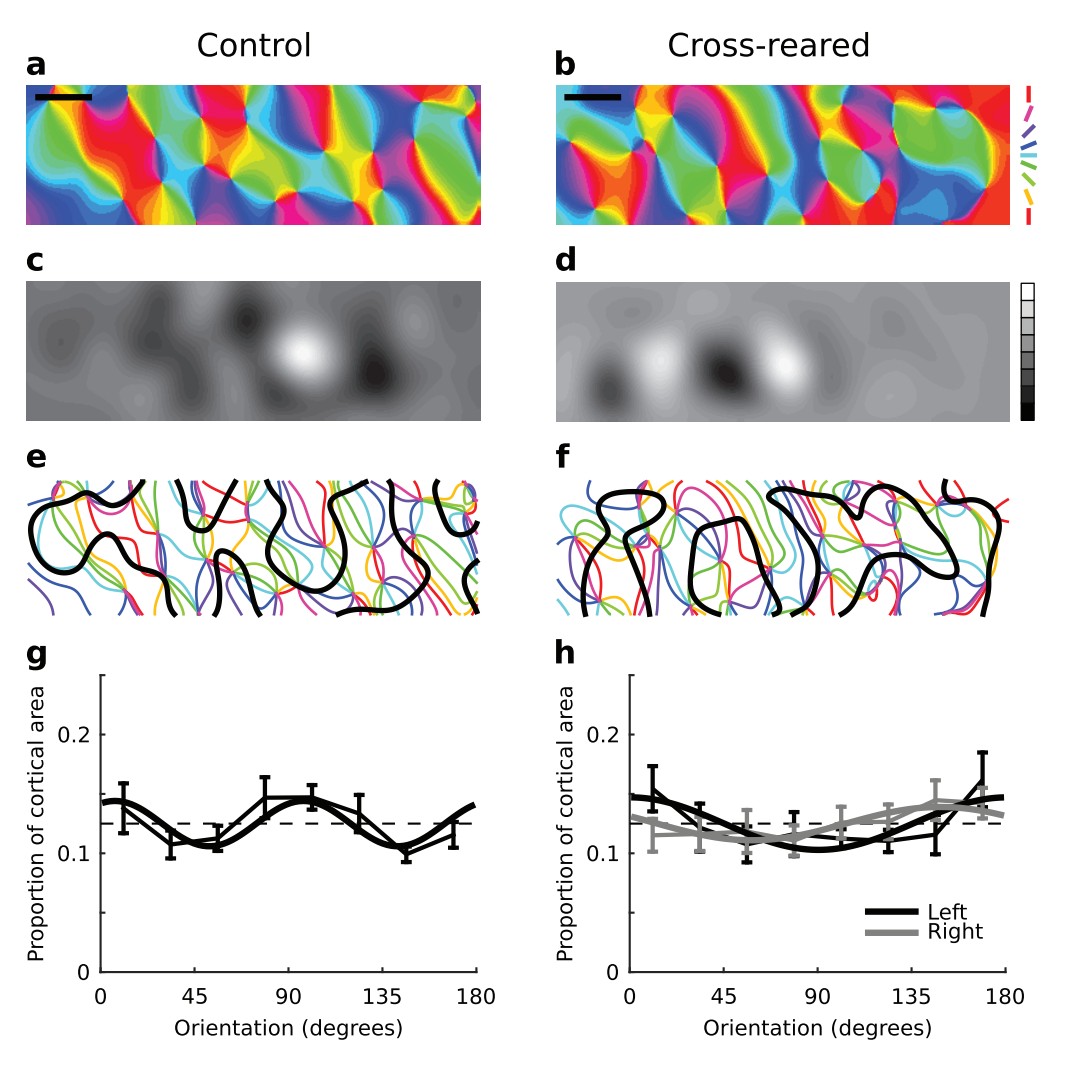

**Figure 5.** Cross-rearing changes the distribution of orientation preferences. (**a**) OP map, (**c**) OD map and (**e**) overlay of OD and OP contours for a control cat. (**b**) OP map, (**d**) OD map and (**f**) overlay of OD and OP contours for a cross-reared cat. While qualitatively the control and cross-reared maps look similar, quantitative analysis revealed differences. (**g**) Proportion of cortical area representing different orientations from binocular stimulation for all control hemispheres (thin line: mean ± 1 SEM, thick line: least-squares sine curve fit). The best-fitting sine curve with period 90° had peaks at 7° and 97°, and an $r^2$ value of 0.6. For comparison the dashed line at a frequency of 1/8 represents equal proportions. (**h**) Data from left (thin black line) and right (thin grey line) monocular stimulation for all cross-reared hemispheres. The best-fitting sine curve with period 180° peaked at 0° for the left eye (horizontal orientations, thick black line) and 148° for the right eye (vertical orientations, thick grey line). The $r^2$ values for the fits were 0.64 (left eye) and 0.71 (right eye). In contrast the best-fitting sine curves with period 90° (not shown) had $r^2$ values of 0.33 (left eye) and 0.13 (right eye). Thus cross-rearing caused a systematic shift in the proportions of the maps occupied by each orientation, towards the orientation that each eye predominantly experienced. As in *Figure 4*, colour encodes the preferred orientation in the OP maps and brightness encodes eye preference in the OD maps. Scale bars: 1 mm. Source data for this figure are available in *Figure 5—source data 1*.

The following source data and figure supplements are available for figure 5:

**Source data 1.** This HDF5 file contains the numerical values shown in *Figure 5*.

**Figure supplement 1.** The distribution of orientation selectivity is slightly altered by cross-rearing.

**Figure supplement 2.** Distributions of orientation preferences in each hemisphere.

cardinal (horizontal and vertical) orientations in the OP map. This is consistent with the distribution of orientation preferences of single units described above, and with previous reports for ferrets (*Coppola et al., 1998*) and cats (*Li et al., 2003*). The proportion of the map devoted to different orientations was well fit by a sine curve with period 90° ($r^2$ = 0.6; *Figure 5g*; *Figure 5—figure supplement 2*). However, in the cross-reared animals the OP maps calculated for each eye showed proportions that were now better fit by sine curves with period 180° ($r^2$ = 0.64, left eye; $r^2$ = 0.71, right eye). In each case these curves peaked close to the orientation of the lens covering that eye (*Figure 5h*; *Figure 5—figure supplement 2*). As a comparison, the best-fitting sine curves with period 90° had $r^2$ values of 0.33 (left eye) and 0.13 (right eye). Thus, cross-rearing caused substantial shifts in the distribution of orientations across the cortex.

## Cross-rearing alters the spatial relationship between OP and OD maps

The characteristic relationships between OP and OD maps are intersection angles and the distance of pinwheels to OD borders. There was no statistically significant change in pinwheel density (*Kaschube et al., 2010*) between rearing conditions (*Figure 6a*). The distribution of intersection angles between OD and OP maps was also similar in both rearing conditions (*Figure 6—figure supplement 1*). There were subtle changes in the spatial distribution of orientation selectivity: in control animals orientation selectivity was slightly greater near OD borders, while in cross-reared animals selectivity was slightly greater near the centre of OD regions (*Figure 6—figure supplement 2*). However there was a clear effect on the distance of pinwheels to OD borders. In our control animals, the distribution of pinwheels relative to OD borders was very similar to that of *Hübener et al. (1997)* (*Figure 6b*) but in cross-reared animals there was a significant shift of pinwheels away from the centre of OD regions (*Figure 6c*). Specifically, there was a significant under representation of pinwheels in the bin corresponding to the centre of OD columns (p=0.01, two-tailed t-test, power = 0.77, 95% confidence interval for difference in means = [0.06, 0.36]), as predicted in our model simulations (*Figure 6d*). Thus, the altered visual input during cross-rearing changed a fundamental aspect of the spatial relationship between OP and OD maps. The similarity between the model prediction and our data provides further evidence that models based on dimension reduction, such as the elastic net, capture essential elements of the mechanisms by which cortical maps develop.

## Discussion

Despite the apparent robustness of the relationship between pinwheels and OD columns in earlier studies that have manipulated visual input, here we showed that this relationship could be altered by cross-rearing. We found consistent changes in cortical response properties from both our electrophysiology and optical imaging. Our findings agree with previous cell population surveys and imaging experiments on cats raised in environments with a single orientation (stripe-rearing) (*Blakemore and Cooper, 1970*; *Sengpiel et al., 1999*) or with orthogonal orientations presented to the left and right eye (*Hirsch and Spinelli, 1970*; *Blakemore, 1976*). These studies, like our own, revealed an over-representation of single units tuned to the orientation that matches the rearing conditions. Interestingly, we found no significant difference in ΔOP – the inter-ocular difference in preferred orientation – of units from our control and cross-reared animals. This might seem surprising given the mismatched inputs in the cross-reared animals. However, previous work using much stronger orientation biases found that cells tend to become monocular rather than maintain large ΔOP values (*Hirsch and Spinelli, 1970*; *1971*; *Blakemore, 1976*). Consistent with these earlier reports, we observed an increase in the monocularity of units from our cross-reared animals compared to the controls (median monocularity index, *MI*, of 0.38 versus 0.24 for the cross-reared and control animals, respectively). However, only about 10% of single units in our cross-reared animals were monocular, suggesting that relatively normal binocular summation developed in these animals. This is consistent with the relatively low power lenses that we used during rearing (-10 dioptres), compared to the higher power lenses used by others (67 dioptres, cats, *Tanaka et al. (2006)*; 167 dioptres, mice, *Kreile et al. (2011)*). Evidently, cortical neurons in our cross-reared animals retained access to a sufficiently broad range of orientations to form fused binocular images during development. Similarly, units from our cross-reared animals generated similar maximum spike rates and had indistinguishable contrast sensitivity to the control animals, again suggesting that the lenses had little impact on basic visual processing.

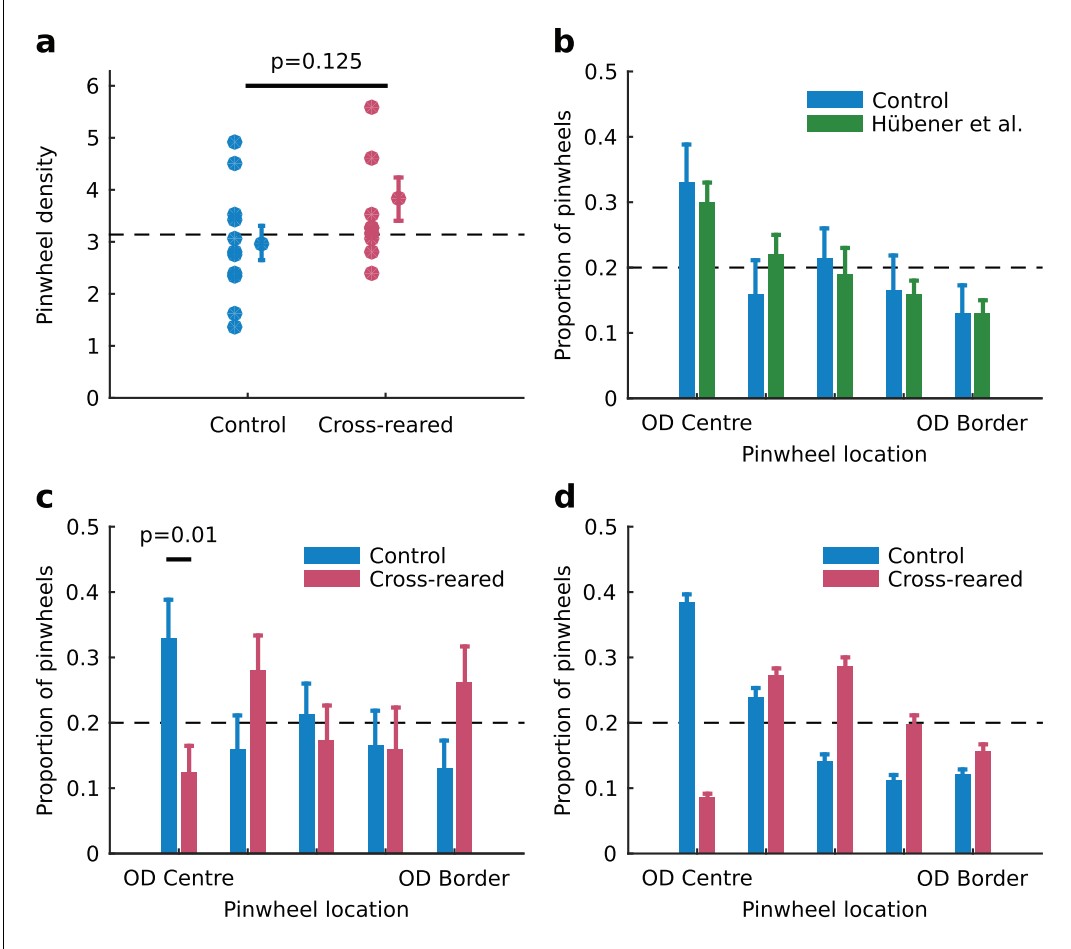

**Figure 6.** Spatial relationship between pinwheels and ocular dominance is modified by rearing condition. (**a**) Pinwheel density relative to squared map wavelength was not significantly different between control and cross-reared animals, both being consistent with the theoretically predicted value of π (dashed line) (***Kaschube et al., 2010***). (**b**) Pinwheel locations relative to the centres/borders of OD regions were quantised into 5 bins similarly to ***Hübener et al. (1997)***. For control animals, pinwheels were disproportionately overrepresented at the centre of OD regions (n = 71 pinwheels total in control hemispheres), consistent with previous data (***Hübener et al., 1997***). (**c**) In strong contrast, for the cross-reared animals pinwheels were disproportionately underrepresented at the centre of OD regions (n = 55 pinwheels total in cross-reared hemispheres). (**d**) Computational simulations using the elastic net reproduced the shift of pinwheels away from the centres of OD regions in the cross-reared compared to control condition (data replotted from ***Figure 1b,f***). For all graphs error bars show ± 1 SEM. p-values in (**a**) and (**c**) are from two-tailed, two-sample t-tests. Source data for this figure are available in ***Figure 6—source data 1***.

The following source data and figure supplements are available for figure 6:

**Source data 1.** This HDF5 file contains the numerical values shown in ***Figure 6***.

**Figure supplement 1.** The distribution of intersection angles of the contours of the OP and OD maps is unchanged by cross-rearing.

**Figure supplement 2.** The spatial layout of orientation selectivity is very slightly altered by cross-rearing.

A theoretical prediction based on symmetry principles was that pinwheels move during normal development and annihilate in pairs (***Wolf and Geisel, 1998***), leading to a final pinwheel density relative to map wavelength of π (***Kaschube et al., 2010***). However chronic imaging experiments have suggested that, with normal visual input, the positions of pinwheels do not move from the positions

at which they are first observed during development (*Chapman et al., 1996*; *Crair et al., 1998*). In adult cats, chronic imaging of OP maps in response to spatially synchronous cortical activity induced by intracortical electrical microstimulation has shown substantial rearrangement of pinwheel positions (*Godde et al., 2002*). However, our results reveal for the first time that pinwheels can be repositioned relative to OD columns purely by abnormal sensory input.

In simulations of normal rearing, pinwheels form near the centre of OD columns to provide good coverage (*Figure 1a,b*): the gradient of OP is highest where the gradient of OD is lowest. However in the cross-reared case the over-represented orientations are the first to appear in the cortex, aligned with the initial formation of OD columns (*Giacomantonio et al., 2010*). The representation of other orientations then forms around this initial configuration, and thus pinwheels are pushed away from the centre of the OD regions, which are dominated by the over-represented orientations (*Figure 1c–f*). In the model, movement of both pinwheels and OD borders contribute to the change in their spatial relationship; however we cannot directly assess whether this is also true in our experimental data.

The degree of plasticity of the tuning of V1 neurons depends on their position in the OP map, in particular whether they lie close to a pinwheel or in an iso-orientation domain (i.e. depending on the spatial gradient of OP). However, the direction of this dependence is controversial: while *Dragoi et al. (2001)* reported greater plasticity of tuning curves at pinwheels in response to an adapting visual input, *Schuett et al. (2001)* found less plasticity at pinwheels in response to pairing a visual stimulus at a particular orientation with direct cortical electrical stimulation. It would be interesting to perform cortical microstimulation studies in cross-reared animals, to determine whether the plasticity of pinwheels that have been repositioned away from the center of OD regions is the same as unperturbed pinwheels in normally reared animals.

From a methodological perspective, we found that blood vessel artefacts common in intrinsic signal imaging of cortex with green light can be overcome by using more statistically sophisticated analysis techniques than are typically employed. Changes in cortical reflectance of green wavelengths are believed to indicate changes in blood volume, while those at longer (red) wavelength primarily indicate changes in blood oxygenation (*Sirotin and Das, 2009*; *Sirotin et al., 2009*). Several previous studies have employed shorter wavelengths for intrinsic signal imaging (*Spitzer et al., 2001*; *Versnel et al., 2002*; *Sirotin and Das, 2009*; *Sirotin et al., 2009*) and our results suggest that this does not strongly affect the resulting maps, provided appropriate analysis techniques are used (*Figure 4*).

Notably, the strength of the lenses worn by our cross-reared animals (-10 dioptres) was considerably less than that in recent stripe rearing studies in cats (67 dioptres) (*Tanaka et al., 2006*) and mice (167 dioptres) (*Kreile et al., 2011*). This was motivated by our preliminary observations that weak lenses were required to avoid obvious behavioural changes in the kittens. The relatively low power of the lenses in our experiment suggests sensitivity of visual cortical structure to relatively small variations in sensory input. This raises the intriguing possibility that a component of the large variability in visual map structure between individuals, seen in cats (*Kaschube et al., 2002*) and humans (*Adams et al., 2007*), could be partly due to individual variations in visual experience (astigmatism, the closest analogy to rearing with cylindrical lenses in humans, has been reported with powers of up to 6 dioptres [*Mitchell et al., 1973*]). More generally, our results redefine the limits of cortical plasticity, and emphasize the importance of appropriate patterns of sensory stimulation during critical periods for normal brain development.

## Materials and methods

All experimental and surgical procedures were approved by the Animal Ethics Committee at the University of Melbourne and were performed in compliance with the Australian Code of Practice for the Care and Use of Animals for Scientific Purposes from the National Health and Medical Research Council of Australia.

### Cat rearing

Five animals wore -10 dioptre cylindrical plano-concave lenses mounted in soft neoprene rubber masks (*Dzioba et al., 1986*). The lenses were fitted to the masks with the planar surface facing towards the cat's eyes. The axis of the cylindrical component for the left eye was vertical and for the

right eye was horizontal. *Figure 2a* shows a pattern of concentric circles (spatial frequency 1 cycle/°). *Figure 2b* shows the same pattern viewed through a -10 dioptre cylindrical lens with its axis oriented horizontally. Both images were captured using a Nikon D90 DSLR camera fitted with a Nikkor 50 mm f/1.2 lens. The cylindrical lens compresses the image along the direction orthogonal to its cylindrical axis. *Figures 2c and d* show the distribution of power over spatial frequency for images of the test pattern viewed normally (*Figure 2c*) and viewed through the -10 dioptre cylindrical lens (*Figure 2d*).

Animals were housed indoors in windowless rooms measuring 2.5 m × 3.0 m. Cross-reared animals wore the masks for 6 hr each day from 3 weeks of age. For the remaining 18 hr each day (i.e., when not wearing the masks) the room lights were extinguished. We took special care to eliminate any stray light entering the room around the doorframe. Six control animals (i.e., not cross-reared) were housed under identical conditions, subject to the same 6:18 hr light:dark cycle. The walls of the rooms housing both control and cross-reared animals were covered with wallpaper consisting of high-contrast images containing all orientations.

## Anaesthesia and surgical procedures

We performed optical intrinsic signal imaging of primary visual cortex in eleven adult cats (3.3–3.8 kg; 6 control, 5 cross-reared). Animals were prepared for acute physiological recordings as described previously (*van Kleef et al., 2010*). Animals were anaesthetised by intramuscular injection of ketamine hydrochloride (10 mg.kg$^{-1}$) and medetomidine (15 µg.kg$^{-1}$). Once deeply anaesthetised, as confirmed by the absence of corneal and toe withdrawal reflexes, animals were placed on a heated surgery table and intubated to ensure adequate respiration. Anaesthesia was then maintained for the remainder of the surgery by inhalation of gaseous isofluorane (0.7–1.0% in $O_2$).

To ensure that adequate levels of oxygenation and anaesthesia were maintained at all times (during surgery and throughout the remainder of the experiment), animals were immediately instrumented to allow continuous monitoring of non-invasive physiological indicators, including heart rate and saturation of peripheral oxygen ($SpO_2$), blood pressure, the electroencephalogram (EEG) and end-tidal $CO_2$ concentration. The EEG was sampled continuously at 256 Hz and Fourier analysis was performed within a sliding window 30 s in duration. The depth of anaesthesia was considered adequate when the power in the EEG was concentrated in the delta band (<4 Hz) and no change in power distribution was observed in response to noxious stimuli. An increase in power in the EEG at frequencies >8 Hz was interpreted as a sign the depth of anaesthesia may be inadequate. If at any stage the depth of anaesthesia was deemed to be inadequate the concentration of the anaesthetic agent was increased.

The cephalic vein was cannulated to permit delivery of fluids and intravenous drugs. Animals were then transferred to a stereotaxic frame and the head fixed using ear bars, a bite bar and a screw placed on the skull at the midline 30 mm anterior to inter-aural zero. Body temperature was maintained at 37.7°C by means of an electric blanket under feedback control.

A craniotomy (10 mm × 15 mm; A4 to P6) was made spanning the midline to expose primary visual cortex (Areas 17 and 18) in both hemispheres. A stainless steel chamber was fixed to the cranium with dental acrylic. The dura mater was removed and the recording chamber filled with silicone oil (Dow Corning 200, 50 cSt) and sealed with a glass cover slip.

To prevent eye movements during recording, animals were subject to neuro-muscular blockade by continuous intravenous infusion of vecuronium bromide (0.1 mg.kg$^{-1}$.h$^{-1}$). During neuro-muscular blockade animals were mechanically ventilated to maintain end-tidal $CO_2$ between 3.5 and 4%. For fluid replacement all animals received a constant intravenous infusion of Hartmann's solution (25% by volume), 5% glucose in 0.9% NaCl solution (25% by volume) and a 10% amino acid solution (Synthamine-17; 50% by volume) at a rate of 2.5 mL.kg$^{-1}$.h$^{-1}$. Animals also received daily injections of atropine (0.05 mg.kg$^{-1}$; s.c.) to reduce salivation, dexamethasone phosphate (1.5 mg.kg$^{-1}$; i.m.) to reduce cerebral oedema, and a broad-spectrum antibiotic (Clavulox; 0.2 mL.kg$^{-1}$; i.m.) to prevent infection.

To prevent desiccation of the corneas the eyes were fitted with neutral power rigid gas-permeable contact lenses. Refractive errors were assessed by reverse ophthalmoscopy and corrected as required using spherical lenses placed in front of the eyes to focus the stimulus on the retina. Eye drops (1% atropine sulphate; 10% phenylephrine hydrochloride) were administered daily to cause dilation of the pupils and retraction of the nictitating membranes.

After surgery and for the remainder of the experiment, anaesthesia was maintained by inhalation of gaseous halothane (0.5–1.0%) in a 60:40 mixture of $N_2O$ and $O_2$. At the conclusion of the experiment animals were euthanized by intravenous injection of an overdose of barbiturate (sodium pentobarbital; 150 mg.kg$^{-1}$).

## Visual stimuli

Visual stimuli were generated by a ViSaGe visual stimulus generator (Cambridge Research Systems Ltd., Cambridge, UK) and displayed on a calibrated Clinton Monoray CRT monitor (modified Richardson Electronics MR2000HB-MED CRT with fast DP104 phosphor, 100 Hz refresh, resolution 1024 × 768 pixels) viewed binocularly or monocularly from a distance of 28 cm.

For imaging, stimuli consisted of luminance defined oriented square wave gratings presented within a circular aperture (diameter 60°) on an isoluminant grey background matched to the mean luminance of the gratings. Orientation preference maps were obtained by recording responses to presentation of high contrast (Michelson contrast, 100%) gratings (0.15 cycles per degree) drifting (temporal frequency 2 Hz) in one of 16 directions equally spaced between 0° and 360°. Each stimulus direction, together with a blank condition (no grating), was presented at least 30 times with the order of presentation randomised across trials.

For single unit recordings, stimuli consisted of patches of sine wave gratings presented within a circular aperture on an isoluminant grey background matched to the mean lumunance of the gratings. For each recorded unit we systematically identified optimal grating parameters (orientation, direction of motion, spatial and temporal frequency, size and position) for stimuli presented monocularly to the right and left eye.

## Intrinsic signal optical imaging

The exposed cortex was imaged using a Pantera 1M60P high-sensitivity 12-bit area scan CCD camera (Teledyne DALSA, Waterloo, ON Canada) fitted with a macroscope consisting of a pair of Nikkor 50 mm f/1.2 lenses (*Ratzlaff and Grinvald, 1991*). The focal plane was positioned 500 μm below the cortical surface, determined by focusing on the surface vasculature and then lowering the camera by way of a micromanipulator. The camera was configured to bin sensor pixels 2×2 producing images with a resolution of 512 × 512 pixels (each pixel being 24 μm square). Image acquisition was restricted to a region of interest comprising a subset of the full imaging frame.

During imaging, the cortex was epi-illuminated using a custom built LED light source with a peak wavelength of 520 nm (Agilent Technologies; HSMQ-C150). Local increases in cerebral blood flow, indicative of neural activity, caused a reduction in cortical reflectance and a corresponding reduction in the resulting image intensity. Cortical responses were imaged during presentation of an ensemble of visual stimuli (gratings, described above). For each stimulus presentation images were acquired continuously at a rate of 5 Hz for a period of 10 s, the onset of which was synchronized to the phase (maximum inspiration) of the respirator. Visual stimuli were presented for 5 s beginning 2 s after the beginning of image acquisition. Each stimulus presentation was followed by a recovery period (minimum duration 3 s) during which the stimulus monitor displayed an isoluminant gray screen, the luminance of which was matched to the mean luminance of the gratings.

## Map generation

All data processing and analyses were conducted using MATLAB. MATLAB code for map pre-processing, generation and analysis is available at https://github.com/nickjhughes/feature-map-stats. Each raw image encompassed both hemispheres, and was cropped to two rectangular regions, one covering the exposed region of areas 17 and 18 in each hemisphere. In cats, both areas 17 and 18 receive direct input from the dorsal lateral geniculate nucleus and are therefore both primary visual cortex (*Payne and Peters, 2001*). The remaining analyses were performed on each hemisphere separately, hereafter a 'dataset.' All of the images in each dataset were then spatially aligned. The first image in the dataset was arbitrarily chosen as the reference frame, and every other image was translated to maximise the linear correlation between it and the reference frame. Trials from opposite directions of stimulus motion were pooled, and then all trials for each stimulus condition for each dataset were pixel-wise averaged, resulting in 50 imaging frames for each of 8 stimulus orientations (0°, 22.5°, . . ., 157.5°), for both eyes. Each of these frames was then high-pass Gaussian filtered (σ =

20 pixels = 480 µm) to remove large scale changes in illumination across the images, and low-pass Gaussian filtered (σ = 2 pixels = 48 µm) to satisfy the requirement of the extended spatial decorrelation method (see below) that the sources be smooth. Image frames were zero-padded prior to filtering. This had no effect on the map regions subsequently analysed since they were sufficiently far from the edges of the frames. The sign of the values in all images were then reversed, as a decrease in reflectance indicates an increase in neural activity.

To generate feature maps from the data we initially tried vector-averaging methods typically used for red-light data (*Hübener et al., 1997*). While these produced OP maps similar to those we subsequently produced using the extended spatial decorrelation (ESD) method of *Stetter et al. (2000)* (see below), the OD maps produced were compromised by blood vessel artefacts. We therefore employed the ESD method for our analysis as follows. Each map (i.e., the two vector components of the orientation preference (OP) map and the ocular dominance (OD) map) was constructed by combining the responses to the appropriate stimulus conditions. Let $R_{\theta,L}$ be the overall average response to orientation $\theta$ produced by the left eye, and similarly for the right eye:

$$
\begin{aligned}
\mathrm{real}(OP) &= \sum_{\theta} 0.5 \big(R_{\theta,L} + R_{\theta,R}\big) \cos(2\theta) \\
\mathrm{imag}(OP) &= \sum_{\theta} 0.5 \big(R_{\theta,L} + R_{\theta,R}\big) \sin(2\theta) \\
OD &= \sum_{\theta} R_{\theta,R} - \sum_{\theta} R_{\theta,L}
\end{aligned}
\tag{1}
$$

where $\theta$ values in the sum were 0°, 22.5°, …, 157.5°. Monocular OP maps were generated similarly, except that only the left or right eye response was considered, rather than the average of the two.

Following the process described by *Stetter et al. (2000)*, these maps were generated using each of the 50 imaging frames separately, resulting in maps that consisted of 50 frames. The frames of each map were first decorrelated from each other using principal component analysis (i.e., each frame's variance was made equal to 1 and the covariance, between frames, was made equal to 0). Single-shift ESD was then performed on each map using a shift of Δr = (5,5) pixels, resulting in 50 sources. The particular shift chosen had no appreciable difference on the resulting maps. The time series of the coefficients of each source were used to choose which of the 50 sources corresponded to the feature map in each case. The source chosen was the one with the coefficient time series that most closely matched that of the typical intrinsic signal optical imaging response (i.e., a rise at the time of stimulus onset, up to a maximal value by the time the stimulus was turned off). The correct source was obvious in most cases (for an example see *Figures 4b–d*). In some cases, the correct source was chosen by looking at which source had both a response with the same spatial extent as the corresponding OP map and an appropriate coefficient time series. Each frame of the chosen source was then multiplied by its corresponding coefficient, and the feature map was defined as the average over frames 31 to 35 (i.e., the final 5 stimulus frames). Finally, each map was low-pass Gaussian filtered (σ = 12 pixels = 288 µm). These filter parameters were chosen to remove any remaining high-frequency noise (due to remaining blood vessel artifacts) while minimising spatial distortion of the resulting feature maps.

## Map analyses

Only areas inside masks were used in the following analyses. Each analysis mask was defined as the area within the anatomical boundaries of area 17/18, with any areas without strong orientation preference removed. All of the above processing was performed prior to this masking.

For a binocular or monocular orientation map, orientation distributions were generated as follows. Each pixel was binned into 22.5° wide orientation preference bins centred on 11.25°, 33.75°, …, 168.75°, according to its preferred orientation. These bins were used to generate a histogram, which we refer to as the orientation distribution. Sine curves were fit to the mean of the histograms for a particular condition (binocular orientation maps from control cats, and left and right monocular orientation maps from cross-reared cats) using least-squares estimation. The period was set to 90° for control maps, and 180° for cross-reared maps, matching the clear and expected periodicity of the data. Fits to the cross-reared maps of sine curves with period 90° were also performed for comparison. The estimated value of the phase of the sine curve was then used to determine the location of the maxima. The quality of the fits was assessed using $r^2$, the square of the correlation coefficient of the data and the fitted function.

To calculate the crossing angle distributions, the zero-level contour of the OD map was calculated, as were the 0°, 22.5°, ..., 157.5° contours of the OP map, and the angle between the OD and OP contours at all points of intersection were calculated. The angle was defined as the difference between the orientations of the tangents of the two contours at the intersection point. The distributions of these angles were calculated for all orientation maps. We compared these between rearing conditions, and also against the expected distribution for unrelated maps, which follows a sine curve (*Morton, 1966*).

Pinwheel locations were defined in orientation maps as described previously (*Carreira-Perpinan et al., 2005*): an integral of orientation preference was calculated along a closed path of radius 1 pixel around each pixel, and the pixel was defined as a pinwheel if the integral was ± 180°. For eight-connected clusters of such pixels, the cluster's centre of mass was defined as the pinwheel location. Pinwheel density was calculated as described previously (*Kaschube et al., 2010*): the number of pinwheels per pixel multiplied by the square of the map wavelength. The wavelength was defined as the mean Fourier wavelength of the map averaged over all directions.

To measure the relationship between the location of pinwheels and the OD domains we used the metric described by *Hübener et al. (1997)* with a slight modification. The positive and negative values in each OD map were separately binned into 5 equally sized regions, corresponding to values ranging from the centres to the borders of the OD columns, and the bin corresponding to each pinwheel was calculated. The bins corresponding to both the positive and negative sections were pooled, leaving 5 bins corresponding to values from the centre to the border of the OD columns. Thus, the centre of the OD column falls into the bin below the 20$^{th}$ percentile and the border region into the bin above the 80$^{th}$ percentile. This metric differs slightly from that used by *Hübener et al. (1997)* in that it considers the regions on either side of 0 separately, rather than binning all the values into 10 equal bins. This modification respects OD borders and takes into account contralateral bias in the OD map.

## Computational modeling

Elastic net simulations were performed as described previously (*Carreira-Perpinan et al., 2005*; *Giacomantonio et al., 2010*). A MATLAB implementation of the elastic net algorithm is available at http://faculty.ucmerced.edu/mcarreira-perpinan/research/EN.html. Parameters for our simulations were as follows. Feature dimensions: 20×20 spatial positions in a square array in a unit square, 2 OD values of -0.05 and 0.05 and 6 OP values of radius 0.08 for each spatial position, giving 4800 feature points in total. Cortical array: 128×128 with non-periodic boundary conditions, α (weighting of coverage term) = 1 for non-overrepresented orientations, α > 1 for the overrepresented orientation in each eye, and β (weighting of continuity term) = 10. Initial value of annealing parameter K = 0.2, multiplied by 0.9925 at each iteration. Simulations (n = 10 per condition) were terminated at K = 0.0358, shortly after the OD and OP maps had formed. The only difference in parameters between simulations of the control and cross-reared conditions were the values of α.

## Single unit recording

After intrinsic signal imaging, extracellular signals from single units were acquired using gold or platinum tipped, lacquer coated tungsten microelectrodes (FHC, Bowdoin, ME USA). The extracellular potential was amplified, band-pass filtered (300 Hz – 5 kHz) and then sampled at 40 kHz using a 1401Plus and Spike2 software (Cambridge Electronic Designs, Cambridge, UK). After isolating a single unit (based on the size and shape of the extracellular spike waveform) the approximate size and location of the receptive fields for the right and left eye were mapped using hand-driven bright or dark bars projected onto a tangent screen. The size and location of the receptive fields together with the unit's tuning for direction, spatial and temporal frequency, and it's sensitivity to stimulus contrast were then determined quantitatively using circular patches of sine wave gratings under the control of the stimulus computer. The receptive fields of all recorded units were located within 10° of the area centralis.

## Analysis of single unit responses

### Analysis of orientation tuning

To quantify the orientation preference and selectivity of each unit we measured responses to 100% contrast drifting sine wave gratings of the preferred spatial and temporal frequency, presented at the optimal size and location, drifting in each of 16 directions equally spaced between 0 and 360°. Responses to opposite directions were averaged together and the set of responses, $R(\theta)$, for orientations ($\theta$) between 0 and 180° were fit with a von Mises function defined as follows:

$$R(\theta) = R_p \exp\big(k\big(\cos\big(2(\theta - \theta_p)\big) - 1\big)\big) + R_0 \tag{2}$$

where $R_p$ denotes the response at the preferred orientation, $\theta_p$, $R_0$ denotes the spontaneous baseline, and $k$ denotes a width parameter from which we calculated the orientation tuning bandwidth. Across all units the goodness of fit, $r^2$, of *Equation 2* to the data formed a distribution with a mean of 0.88 and median of 0.93.

### Analysis of spatial frequency tuning

To quantify the preferred spatial frequency and the spatial frequency tuning bandwidth we measured responses to 100% contrast drifting sine wave gratings of the preferred orientation, presented at the optimal size and location, with a range of frequencies ($f$) from 0.05 to 1.6 cycles per degree. The responses, $R(f)$, were fit with a skewed Gaussian function defined as follows:

$$R(f) = R_p \exp\left(-\left(\frac{\log\left(\frac{f}{f_p}\right)}{k + \lambda \log\left(\frac{f}{f_p}\right)}\right)^2\right) + R_0 \tag{3}$$

where $R_p$ denotes the maximal response at the preferred frequency, $f_p$, $k$ controls the tuning bandwidth, $\lambda$ controls the skew of the tuning curve and $R_0$ denotes the spontaneous baseline. Across all units, the goodness of fit, $r^2$, of *Equation 3* to the spatial frequency tuning data formed a distribution with a mean of 0.89 and a median of 0.96.

### Analysis of temporal frequency tuning

To quantify the preferred temporal frequency and temporal frequency tuning bandwidth we measured responses to 100% contrast drifting sine wave gratings of the preferred orientation and spatial frequency, presented at the optimal size and location, with a range of temporal frequencies from 0.25 to 24 Hz. As for the analysis of spatial frequency tuning, the responses over the full range of temporal frequencies were fit with a skewed Gaussian function as defined in *Equation 3* (with $f$ denoting temporal rather than spatial frequency). Across all units, the goodness of fit, $r^2$, of *Equation 3* to the temporal frequency tuning data formed a distribution with a mean of 0.86 and a median of 0.93.

### Analysis of contrast sensitivity

To quantify the contrast sensitivity of each unit we measured responses to optimal drifting sine wave gratings presented at Michelson contrasts ($c$) ranging from 1 to 100%. The set of responses, $R(c)$, were fit with a sigmoid function defined as follows:

$$R(c) = R_m \frac{c^n}{c^n + \sigma^n} + R_0 \tag{4}$$

where $R_m$ denotes the maximal saturated response above the spontaneous baseline, $R_0$, $n$ is an exponent that determines the slope of the curve, and $\sigma$ is the semi-saturation contrast, corresponding to the stimulus contrast at which the response equals half $R_m$. Across all units the goodness of fit, $r^2$, of *Equation 4* to the data formed a distribution with a mean of 0.9 and median of 0.96. Units that did not show response saturation at higher contrasts were excluded from the analysis.

## Analysis of ocular dominance

To quantify the ocular dominance of each unit we computed a monocularity index, defined as follows:

$$MI = \frac{|R_R - R_L|}{R_R + R_L} \qquad (5)$$

where $R_R$ and $R_L$ denote the response to a patch of the optimal sine wave grating presented monocularly to the right and left eye, respectively.

## Acknowledgements

This work was supported by the Australian Research Council (ARC) Centre of Excellence in Vision Science (CE0561903) and the Centre of Excellence for Integrative Brain Function (CE140100007), the National Health and Medical Research Council of Australia (GNT0525459), and by the Victorian Lions Foundation. We thank Molis Yunzab for help with animal rearing and with the experiments and Jonathan J. Hunt for his contributions to this work.

## Additional information

### Funding

| Funder | Grant reference number | Author |
|---|---|---|
| Victorian Lions Foundation | Lions Vision Research Fellowship | Shaun L Cloherty |
| National Health and Medical Research Council | GNT0525459 | Geoffrey J Goodhill Michael R Ibbotson |
| Australian Research Council | CE140100007 | Michael R Ibbotson |
| Australian Research Council | CE0561903 | Michael R Ibbotson |

The funders had no role in study design, data collection and interpretation, or the decision to submit the work for publication.

### Author contributions

SLC, MAH, Acquisition of data, Analysis and interpretation of data, Drafting or revising the article; NJH, Analysis and interpretation of data, Drafting or revising the article; PSB, Acquisition of data, Drafting or revising the article; GJG, Conception and design, Analysis and interpretation of data, Drafting or revising the article; MRI, Conception and design, Acquisition of data, Analysis and interpretation of data, Drafting or revising the article

### Author ORCIDs

Shaun L Cloherty, http://orcid.org/0000-0002-7679-1764
Nicholas J Hughes, http://orcid.org/0000-0003-1657-4378
Geoffrey J Goodhill, http://orcid.org/0000-0001-9789-9355
Michael R Ibbotson, http://orcid.org/0000-0002-3803-6653

### Ethics

Animal experimentation: All experimental and surgical procedures were approved by the Animal Ethics Committee at the University of Melbourne (Ethics ID: 1112238.1) and were performed in compliance with the Australian Code of Practice for the Care and Use of Animals for Scientific Purposes from the National Health and Medical Research Council of Australia.

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
