## [Decision Letter]

Thank you for submitting your work entitled "Sensory experience modifies feature map relationships in visual cortex" for consideration by *eLife*. Your article has been reviewed by two peer reviewers, and the evaluation has been overseen by a Reviewing Editor and David Van Essen as the Senior Editor.

The reviewers have discussed the reviews with one another and the Reviewing Editor has drafted this decision to help you prepare a revised submission.

Summary:

This paper reports that the relationship between pinwheel position and ocular dominance column boundaries changes in cross-orientation reared cats. Notably, the pinwheels are less concentrated in the centers of the ocular dominance stripes than they are in normal control animals. This result is most clearly documented in Figure 6. The experimental findings are novel and it is also of interest that it is predicted by dimension-reduction models of visual cortex development, notably in prior work by Giacomantonio et al. (2010). Overall the paper adds an interesting new finding to the literature on cortical map plasticity; it also adds usefully to the literature on stripe-rearing experiments in cats which is not extensive, though it has a long and somewhat controversial history.

Essential revisions:

1) Figure 1 shows simulation results with pinwheel positions (black dots) and OD borders (black lines). The positions of the black dots do not seem to correspond at all to the pinwheel locations in panels a), c) or e). There are pinwheels without black dots, and vice versa. I would assume this is a mistake made in preparing the figure. However, in panel e) it is visually very obvious that the pinwheel positions are almost always very close to an OD border. This seems at odds with the histogram in panel f) which shows a rather weak tendency for them to be about halfway between the OD centers and the edges. I might be wrong about this, but it does make me worry that something more serious has gone wrong with the analysis.

2) The authors acknowledge (Discussion, third paragraph) that the experimental results may reflect either pinwheel movement or OD boundary movement (both is clearly also a possibility). However, the results are generally presented as suggesting that it is the pinwheels that move (e.g. Discussion, second paragraph). The authors might try to come up with evidence that pinwheel statistics are different in the cross-reared animals, or some other kind of evidence that might point specifically to pinwheel movement. In the absence of it, I think the paper should be careful to be clear about the ambiguity: it is a slightly less interesting result if it is mainly due to OD border movement. It is also relevant because cross-orientation rearing was thought by Hubel and Wiesel and others to cause selective monocular deprivation rather than to modify orientation preference per se. Arguments against that view might usefully be made in the paper. I presume that the model really does show a shift of pinwheel positions – that might be made clear, if so.

3) The eyes of cats rotate (cyclotorsion) by a few degrees under anesthesia – (e.g. Cooper and Pettigrew, 1979, J. Comp. Neurol., 184, 1-26) resulting in a roughly 15 deg difference between the two eyes. Though this is fairly small in relation to the effects seen here I think it would be worth correcting for. In particular, it ought to be manifest in the data on inter-ocular orientation differences (referred to in the manuscript but values not given). A figure showing the raw data (perhaps in the Supplementary figures) would be useful as well as some actual numbers on, or around, in the third paragraph of the subsection “2Cross-rearing alters tuning properties of single cells”.

4) Although the signal is stronger with green imaging, and blood vessels can be removed, I presume the vessels still obscure or distort the optical signals from the tissue underneath in a way that does not happen with red light. So there may still be some disadvantage to green imaging. I was also concerned that the difference between Figure 4 (green vs. red imaging results) was quite large and especially that many fewer pinwheels were visible with green imaging. This must surely have some effect on the analyses, though what might be hard to say.

5) Related to the previous point, I was also concerned about the large size of the low pass filter employed (at the end of the subsection “Map generation”: 288 microns, or 12 pixels). This will surely remove real structure from the maps, not just noise. Was such severe smoothing really necessary? What do the results look like if you omit this smoothing? I would strongly suggest re-doing the analyses with much less smoothing – even though I know it would probably be a lot of work.

6) Figure 6—figure supplement 1. Buried away here is what seems like an unfortunate finding that intersection angles between OD and OP contours in the experimental animals appear to be random (as well as being the same in the control and experimental groups) inasmuch as the distributions follow the sine function predicted for random curves (Morton, 1966). However, in Hubener et al. (1997) the distribution functions for border intersection angles in unrelated maps were essentially flat (see Figure 5 in that paper). Who is wrong, Morton or Hubener et al.? Are the results in Figure 6—figure supplement 1 actually consistent with those of Hubener et al.?

[Editors' note: further revisions were requested prior to acceptance, as described below.]

Thank you for resubmitting your work entitled "Sensory experience modifies feature map relationships in visual cortex" for further consideration at *eLife*. Your revised article has been favorably evaluated by David Van Essen (Senior editor), a Reviewing editor, and two reviewers.

The manuscript has been improved but there are some remaining issues that need to be addressed before acceptance, as outlined below:

Reviewer #1:

The authors have responded substantively to my comments. However, the newly added Figure 1 appears to be lacking axes. As far as I could tell this was not a local pdf rendering problem.

Reviewer #2:

My main point concerned Figure 1.

For the first figure, it seems indeed that a reflexion of the pinwheel locations along the anti-diagonal produces the new plot. I am now more convinced that the data analysis is correct for this figure.

[Editors' note: further revisions were requested prior to acceptance, as described below.]

While we are almost there, I agree as editor that Figure 1 can be improved.

Part of the confusion might be that on first sight it is unclear whether this is a spatial plot or some random curve.

A scale bar without units but also present in panel a-c would solve this.

Perhaps it is even possible to indicate the zoom region in panel a-c.

In addition, is Figure 2 missing the a and b labels (c and d are fine)?

---

## [Author Response]

Essential revisions:

1) Figure 1 shows simulation results with pinwheel positions (black dots) and OD borders (black lines). The positions of the black dots do not seem to correspond at all to the pinwheel locations in panels a), c) or e). There are pinwheels without black dots, and vice versa. I would assume this is a mistake made in preparing the figure.

Our sincerest apologies – there was indeed an error introduced when plotting the figure, which is that the x and y coordinates of the pinwheels were inadvertently transposed. We have corrected this in the revised figure. This applied only to that figure, and the pinwheel positions were correct for all the quantitative analysis.

However, in panel e) it is visually very obvious that the pinwheel positions are almost always very close to an OD border. This seems at odds with the histogram in panel f) which shows a rather weak tendency for them to be about halfway between the OD centers and the edges. I might be wrong about this, but it does make me worry that something more serious has gone wrong with the analysis.

The results are correct. The metric we used, following Hübener et al. (1997), is defined functionally (in terms of the overall distribution of OD values) rather than geometrically. This overcomes the problem of trying to define a consistent spatial metric for distance between OD borders, which is hard to do when the borders are irregular. With the Hübener metric it is not possible to determine accurately by eye from Figure 1 what the histogram in Figure 1 will look like, since the histogram relies on the full distribution of OD values, which is not represented in Figure 1 (which shows only the borders, i.e. where OD reverses sign).

2) The authors acknowledge (Discussion, third paragraph) that the experimental results may reflect either pinwheel movement or OD boundary movement (both is clearly also a possibility). However, the results are generally presented as suggesting that it is the pinwheels that move (e.g. Discussion, second paragraph). The authors might try to come up with evidence that pinwheel statistics are different in the cross-reared animals, or some other kind of evidence that might point specifically to pinwheel movement. In the absence of it, I think the paper should be careful to be clear about the ambiguity: it is a slightly less interesting result if it is mainly due to OD border movement.

We apologize that we were unclear on this point. For the experimental data it is difficult to definitively establish the relative contributions of pinwheel vs. OD border movement to the altered histograms. However, it is possible to show in the model that both pinwheels and OD borders move, and in response to the reviewer’s comment we have now added these results to the paper (Figure 1). In particular, we fixed the random seed in the algorithm so that the same map was reproducible, and then gradually increased the strength of over-representation induced by cross-rearing (α) across different simulations. This allowed us to track the positions of both pinwheels and OD borders as a function of α. An example is shown in Figure 1: as α is increased both the pinwheel and the nearest OD border move, but the distance between them is reduced. The average movement of pinwheels away from their initial (α = 1) positions in the map as a function of α is shown in Figure 1. We return to these results in the Discussion: `… In the model, movement of both pinwheels and OD borders contribute to the change in their spatial relationship; however we cannot directly assess whether this is also true in our experimental data.’.

It is also relevant because cross-orientation rearing was thought by Hubel and Wiesel and others to cause selective monocular deprivation rather than to modify orientation preference per se. Arguments against that view might usefully be made in the paper. I presume that the model really does show a shift of pinwheel positions – that might be made clear, if so.

We address this in the first paragraph of the Discussion. Specifically, we note that in our cross-reared animals, relatively few cells (only ~10%) were strictly monocular, suggesting that binocular summation was relatively normal in these animals.

3) The eyes of cats rotate (cyclotorsion) by a few degrees under anesthesia – (e.g. Cooper and Pettigrew, 1979, J. Comp. Neurol., 184, 1-26) resulting in a roughly 15 deg difference between the two eyes. Though this is fairly small in relation to the effects seen here I think it would be worth correcting for. In particular, it ought to be manifest in the data on inter-ocular orientation differences (referred to in the manuscript but values not given). A figure showing the raw data (perhaps in the Supplementary figures) would be useful as well as some actual numbers on, or around, in the third paragraph of the subsection “Cross-rearing alters tuning properties of single cells”.

The rotation (inward at the top) between pupils, after paralysis, reported by Cooper and Pettigrew was +8.7 ± 4.7° (their Table I). This is comparable to the mean inter-ocular difference in preferred orientations (ΔOP) we see in our data. We have modified the text to reflect this: ‘…we found a torsional disparity in the preferred orientation of the two eyes in both our control (mean ΔOP = 11.7°, p < 0.001, two-tailed t-test) and cross-reared (mean ΔOP = 9.8°, p < 0.001, two-tailed t-test) animals. …’. We have also added a supplement to Figure 3 (Figure 3—figure supplement 1) showing the distributions of ΔOP for our control and cross-reared animals.

To correct for this cyclotorsion, we would have to add a small angle to orientations for the left eye, and subtract a small angle from orientations for the right eye. One way to do this would be to assume a normative difference of 8.7°, as per Cooper and Pettigrew, and apportion the rotation equally between the two eyes. In that case the angle we would need to add/subtract is 4.35°. However, the angular resolution with which we measured, plotted and compared orientation preference was 22.5°, so such a correction would have no substantive effect on the distributions or the fitted sine curves shown in Figure 3. Given these considerations, we think it most appropriate if we report the data as it was collected rather than risk introducing correctional errors.

4) Although the signal is stronger with green imaging, and blood vessels can be removed, I presume the vessels still obscure or distort the optical signals from the tissue underneath in a way that does not happen with red light. So there may still be some disadvantage to green imaging.

We agree that red light may have some advantages. However, we are confident based on our measurements and analysis that these are outweighed by the far stronger signal offered by green light. In fairness though, we have now deleted our previous strong claim that green light constitutes the best method for cortical intrinsic signal imaging.

I was also concerned that the difference between Figure 4 (green vs. red imaging results) was quite large and especially that many fewer pinwheels were visible with green imaging. This must surely have some effect on the analyses, though what might be hard to say.

Although we agree that there are slightly fewer pinwheels in the example map produced from green light, the overall map structure is very similar. Since we only have n = 1 animal for this comparison it is hard to draw strong conclusions about whether systematic differences exist. However, the key point is that all the imaging in the normal vs. cross-rearing experiment was done with green light, hence we are comparing apples with apples.

5) Related to the previous point, I was also concerned about the large size of the low pass filter employed (at the end of the subsection “Map generation”: 288 microns, or 12 pixels). This will surely remove real structure from the maps, not just noise. Was such severe smoothing really necessary? What do the results look like if you omit this smoothing? I would strongly suggest re-doing the analyses with much less smoothing – even though I know it would probably be a lot of work.

We agree with the reviewer that this is an important point, and so we present the data inFigure 7 which we calculated for the maps using low-pass filters with sizes of 4 pixels and 8 pixels. Overall the maps look very similar, as can be seen from the examples shown (same map as Figure 5 in the manuscript). Given that the maps looked very similar, we chose 12 pixels because it most clearly removed the remaining blood vessel artefacts.

Author response image 1.**DOI:**
http://dx.doi.org/10.7554/eLife.13911.021

6) Figure 6—figure supplement 1. Buried away here is what seems like an unfortunate finding that intersection angles between OD and OP contours in the experimental animals appear to be random (as well as being the same in the control and experimental groups) inasmuch as the distributions follow the sine function predicted for random curves (Morton, 1966). However, in Hubener et al. (1997) the distribution functions for border intersection angles in unrelated maps were essentially flat (see Figure 5 in that paper). Who is wrong, Morton or Hubener et al.? Are the results in Figure 6—figure supplement 1 actually consistent with those of Hubener et al.?

We thank the reviewer for drawing attention to these subtle points. Hübener et al. (1997) calculated their distributions of intersection angles by computing gradient fields for the two maps, and then taking the angular difference between the two gradients at pixels located on the OD borders. For *unrelated* maps this results in a flat distribution, as noted by the reviewer. However, we instead calculated the OD and OP contour lines, and then took the difference between their tangent angles at their points of intersection (as used previously by e.g. Bartfeld and Grinvald (1992) and Swindale (Cerebral Cortex, 2000)). In this case the distribution between *unrelated* maps is a sine curve, as shown analytically by Morton (1966). So both Morton and Hübener et al. are correct with regard to the distributions for *unrelated* maps.

However, our results in Figure 6—figure supplement 1, for *related* maps do appear to be different from Hübener et al., in that we did not detect a non-random relationship (i.e. deviating from a sine curve), for either the control or the cross-reared maps. The reasons for this are unclear. However, since our calculation is based on contour lines rather than pixels the n values in this case are relatively low, and it is apparent in the figure that the SEMs are quite large. Thus, we do not feel that our data in this regard constitute a strong challenge to the results of Hübener et al. (1997). We have added a statement to the figure caption to make this point: `… For these data we cannot say in either case that the distributions are non-random. …’.

[Editors' note: further revisions were requested prior to acceptance, as described below.]

The manuscript has been improved but there are some remaining issues that need to be addressed before acceptance, as outlined below:

Reviewer #1:

The authors have responded substantively to my comments. However, the newly added Figure 1 appears to be lacking axes. As far as I could tell this was not a local pdf rendering problem.

Figure 1 shows a cropped region of a set of simulated OP and OD maps (similar to the complete maps shown in Figure 1). The spatial units in these simulations are arbitrary, and therefore we don’t feel that adding axes is useful. The overall scale of the plot is apparent from the distance between the pinwheels and the OD borders.

Reviewer #2:

My main point concerned Figure 1.

For the first figure, it seems indeed that a reflexion of the pinwheel locations along the anti-diagonal produces the new plot. I am now more convinced that the data analysis is correct for this figure.

We thank the reviewer again for spotting our original error. No further changes have been made to the manuscript or figures.

[Editors' note: further revisions were requested prior to acceptance, as described below.]

While we are almost there, I agree as editor that Figure 1 can be improved.

Part of the confusion might be that on first sight it is unclear whether this is a spatial plot or some random curve.

A scale bar without units but also present in panel a-c would solve this.

Thank you for this good suggestion. In the revised submission we have added scale bars to Figure 1. In all cases the scale bar indicates 15 pixels in the simulated feature maps. We have also modified the figure legend accordingly.

Perhaps it is even possible to indicate the zoom region in panel a-c.

In addition, is Figure 2 missing the a and b labels (c and d are fine)?

Unfortunately we have not been able to reproduce the issue of concern here. The original pdf artwork for Figure 2 includes panel labels for all four panels (A-D). The panels labels appear to be present in both the.pdf file for Figure 2 and in the merged.pdf file downloaded from the *eLife* submission system.